# Visible-Light Activation of Photocatalytic for Reduction of Nitrogen to Ammonia by Introducing Impurity Defect Levels into Nanocrystalline Diamond

**DOI:** 10.3390/ma13204559

**Published:** 2020-10-14

**Authors:** Rui Su, Zhangcheng Liu, Haris Naeem Abbasi, Jinjia Wei, Hongxing Wang

**Affiliations:** Key Lab for Physical Electronics and Devices, Faculty of Electronics and Information Engineering, Xi’an Jiaotong University, Xi’an 710049, China; angelina123@stu.xjtu.edu.cn (R.S.); zcliu_2043@xjtu.edu.cn (Z.L.); haris1996@stu.xjtu.edu.cn (H.N.A.); jjwei@mail.xjtu.edu.cn (J.W.)

**Keywords:** diamond, impurity levels, nitrogen vacancy center, solvated electrons, ammonia synthesis

## Abstract

Nitrogen impurity has been introduced in diamond film to produce a nitrogen vacancy center (NV center) toward the solvated electron-initiated reduction of N_2_ to NH_3_ in liquids, giving rise to extend the wavelength region beyond the diamond’s band. Scanning electron microscopy and X-ray diffraction demonstrate the formation of the nanocrystalline nitrogen-doped diamond with an average diameter of ten nanometers. Raman spectroscopy and PhotoLuminescence (PL) spectrum show characteristics of the NV^0^ and NV^−^ charge states. Measurements of photocatalytic activity using supraband (λ < 225 nm) gap and sub-band gap (λ > 225 nm) excitation show the nitrogen-doped diamond significantly enhanced the ability to reduce N_2_ to NH_3_ compared to the polycrystalline diamond and single crystal diamond (SCD). Our results suggest an important process of internal photoemission, in which electrons are excited from negative charge states into conduction band edges, presenting remarkable photoinitiated electrons under ultraviolet and visible light. Other factors, including transitions between defect levels and processes of reaction, are also discussed. This approach can be especially advantageous to such as N_2_ and CO_2_ that bind only weakly to most surfaces and high energy conditions.

## 1. Introduction

It has been known that the hydrogen terminated diamond, whose conduction-band edge lies 0.8–1.3 eV above the vacuum level, can emit electrons without any barrier when illuminated by light with bandgap above 5.45 eV [1,2,3,4,5,6]. Recently, boron-doped diamond electrodes have been applied to chemistry due to their exceptional properties. For example, they are very stable when used as a large electrochemical potential window and a control of surface termination [7,8,9]. However, photoemission processes of diamond film require high-energy photons due to its large bandgap. Only the light with wavelength less than 225 nm can excite across the 5.45 eV bandgap. However, it is the ultraviolet light with 300–400 nm from solar radiation that can pass through atmosphere to the surface due to ozone [10,11,12]. This leads to the low efficiency for photocatalyst. Therefore, in order to extend wavelength range and promote applications, our work will focus on the research of nitrogen impurity-doped diamond film in the field of photocatalysis. Since impurities can introduce intermediate levels, especially optical active energy levels, in diamond forbidden band.

Ammonia has been widely utilized as an important energy carrier, fertilizer precursor, and fuel because of its high hydrogen density, low liquefying pressure, and carbon-free emission [13,14,15,16]. Despite its importance, production is still heavily dependent on the energy intensive Haber–Bosch process owing to harsh conditions (typically 300–500 °C and 200–300 atm) [17,18,19]. Compared with the Haber–Bosch method, nitrogen fixation of photocatalyst can be promoted at room temperature and pressure with clean and energy-rich solar energy as the driving force. Water and nitrogen as raw materials have a wide range of sources [20,21,22]. In addition, the electrocatalytic nitrogen reduction to synthesize ammonia requires solar, wind, and tidal energy to be converted into electrical energy first, and then the nitrogen fixation process can be realized through the electrocatalytic process [23,24,25]. Therefore, the photocatalytic nitrogen reduction to produce ammonia has a low cost and provides an alternative new method for ammonia synthesis. However, photocatalytic transformations from N_2_ to NH_3_ at semiconductor surfaces are typically limited by adsorption and activation of nitrogen molecules. Owning that the cleavage of strongest bond (N≡N triple bond) is extremely difficult, the critical step on formation of N_2_ + e^−^ + H^+^→N_2_H requires approximately 3 eV of energy [26,27]. Hence, the efficiency of production is still ultralow from N_2_ by the electrosynthesis process.

Here, we reported a strategy to endow a boosting performance on nitrogen reduction reaction (NRR). Nitrogen-doped diamond with negative electron affinity (NEA)as electron source to reduce N_2_ to NH_3_ can be facilitated without any contact between nitrogen and diamond. We also compared performance between nitrogen-doped polycrystalline diamond film (NDD), single crystal diamond (SCD) film, and non-doped polycrystalline diamond (PD) film. The results show that the nitrogen-doped diamond film has a much better performance under sub-band gap illumination, since negative charge state can give rise to photoinitiated electrons, owing that excited states are close to the band edges.

## 2. Materials and Methods

### 2.1. The Growth of Nitrogen-Doped and Undoped Polycrystal Diamond

Two inch Si substrates were cleaned by sonication in the acetone, alcohol, and water about 15 min. The performance of diamond nucleation enhancement stage has a deep impact on the quality of the aforementioned applications of nanocrystalline diamond (NCD) films. It is essential to investigate and optimize this stage, aiming at the high adaptability of the films for further application [28]. Therefore, all samples were seeded by sonication in 0.3 g nanodiamond particles (5–8 nm in diameter, Sigma Aldrich, St. Louis, MO, USA) and 20 mL ethanol for 15 min to achieve a high nucleation density. Then, about 3 μm polycrystalline diamond layers were grown on the samples by microwave plasma chemical vapor deposition (MPCVD) in a modified AsTex system (Carat Systems, Boston, MA, USA). The film thickness was calculated by an average growth rate 20 μm/h which was evaluated from weight of numbers of samples under same growth conditions. The conditions of both nucleation and subsequent growth steps were as follows: total gas flow rate of 500 sccm, CH_4_/H_2_ ratio of 4%, gas pressure of 100 Torr, growth temperature of 1000 °C, and microwave power of 2000 W. After that, 200 nm nitrogen-doped diamond layer was grown by MPCVD on both nitrogen-doped and undoped polycrystal diamond. The deposition parameters kept consistent with that of polycrystalline diamond besides the 1% N_2_/H_2_ ratio.

### 2.2. Growth of Single Crystalline Diamond Film

High pressure and high temperature Ib-type diamond as substrate was utilized to deposition of undoped single crystal diamond homoepitaxial layer. The thickness of film was 500 nm. Compared with deposition conditions of the polycrystalline diamond, the CH_4_/H_2_ ratio, gas pressure, growth temperature, and microwave power were transformed to 5%, 85 Torr, 1000 °C, and 1000 W, respectively. Then, 200 nm nitrogen-doped diamond film was grown on samples. The conditions of subsequent growth steps were identical with that of nitrogen-doped polycrystal diamond films.

### 2.3. The Characterization of Diamond Film

Scanning electron microscopy (SEM) measurements were executed by using Gemini SEM 500 microscope (Carl Zeiss, Munich, Germany). The Raman spectroscopy were obtained by laser Raman spectrometer with excitation at 632 nm. The PL spectrum was performed using Edinburgh FLS9 (LongRun, Xi’an, China) with excitation at 325 nm. Moreover, the UV-Vis-NIR visible spectrum was measured by PE Lambda950 (Cernet, Beijing, China) at a wavelength of 697.5 nm and experiments were set up by photocatalysis system CEL-SPH2N (AuLight, Beijing, China). The surface resistance was obtained by an Agilent B1505A power device analyzer (Agilent Technologies, Santa Clara, CA, USA).

### 2.4. Photoinitiated Reduction of N_2_ to NH_3_

A 450 W high-pressure Hg/Xe lamp was used for photo-induced ammonia production, which was installed in an Oriel lampshade 12 inches from the sample. Wavelength range was from 200 to 800 nm and the reaction vessel was indicated in Figure 1. N_2_ was entered into the gas filter in order to make sure purity of the reaction gas without impurities. Experiments are carried out under sealed condition without mixed gas to ensure the accuracy. Then, slowly blow N_2_ into a container filled with 18.2 MΩ water (Barnsted NanoPure, Thermo Scientific, Waltham, MA, USA) and 0.01 M high purity (Alfa Puratronic, Charlotte, NC, USA, 99.9955%) Na_2_SO_4_. The vessel was sealed with a quartz cover and illuminated for different periods of time. The gas exported from vessel was run through the NH_3_ absorption bottle and finally taken out the system which aimed to detect other generated remaining gas. The ammonia yield from the reaction vessel and NH_3_ absorption bottle was measured by using the indophenol blue method for specific lengths of time [29,30]. Before experiments, a standard absorption curve was obtained using ammonium chloride solutions with known concentration. After completing experiments, we removed a 2 mL aliquot of the solution from the reaction vessel. Then, dilute 0.100 mL of 0.05 mol/L NaClO with 2 mol/L NaOH and 0.1 mL of 0.4% (by weight) Na [Fe(NO)(CN)_5_] (sodium nitroferricyanide) aqueous solution. Finally, 0.5 mL solution of 5% salicylic acid, 5% sodium citrate (by weight), and 2 mol/L NaOH were added to the above solution. All of reagents were utilized in their purest form. The accuracy of the method is that when 1.0, 3.0, 5.0, 7.0 µg ammonia yield is added to the sample solution, the average recovery is 100%. The precision of the method is when the ammonia yield in the sample is 1.0, 5.0, 10.0 µg/10 mL, the coefficient of variation is 3.1%, 2.9%, 1.0%. The average relative deviation is 2.5%. Effective measures have been taken to eliminate interferences. For example, multiple cations (Ca^2+^, Fe^3+^, Mg^2+^, Mn^2+^) have been complexed by salicylic acid. After 1 h, the absorption spectrum was measured using a PE Lambda950 ultraviolet-Vis-NIR spectrophotometer. The formation of indophenol blue was determined using absorbance at a wavelength of 635 nm [31,32].

## 3. Results

The nitrogen-doped nanocrystalline diamond film was produced and consisted of randomly oriented well-faceted grains. As shown in Figure 2a, the appearance looks like clouds cluster. Such porosity can achieve high active site density. When the image is enlarged 50 times (Appendix A), small nanoparticles with an average diameter of more than 10 nanometers can be observed. As depicted in Figure 2b, polycrystalline diamond film with larger grains shows obvious boundary with tens of micro grain diameter. Additionally, there is no cluster spherical shape. For the purpose of preventing the influence of CN species that are on the N-doped diamond surface or in the grain boundaries, single crystalline diamond (SCD) film was studied for a contrast. Clearly, in Figure 2c, the surface of SCD was flat with no grand boundaries. The film thicknesses of all samples were 200 nm. Actually, photocatalytic oxidation activity occurs more easily in grain boundaries. Since, it is weak bonds between grain boundaries that might be broken by the irradiation, resulting in a higher probability of oxygen radicals, such as O_3_. The oxygen radicals are produced by the reaction induced by UV light just at the top of the diamond surface with water during the photo-irradiation process [33,34,35]. In addition, XRD spectrum was illustrated in Appendix A. The crystal plane indices demonstrated existence of the diamond.

Raman spectroscopy is employed for analyzing the composition of chemical vapor deposited diamond films, mainly because the scattering effect of graphite and amorphous carbon (peaks on 1520 cm^−1^) is very obvious and 1332 cm^−1^ has been recognized as a characteristic peak derived from sp3 diamond crystals [11,36]. Wherefore, the 632 nm laser photoexcitation was used to measure Raman spectra of N-doped diamond, and undoped poly-crystalline diamond, as shown in Figure 3a,b, respectively. Peaks on 1332 and 1520 cm^−1^ represent diamond and graphite in Raman spectra of the nitrogen-doped diamond film, which means D (diamond) and G (graphite) bands. Differently, the peak on 500 cm^−1^ as a result of silicon substrate in the polycrystal diamond film is much higher than it is in the nitrogen diamond film. However, silicon has no effect in the experiment, there will be a comparative experiment in the following part. In addition, the characteristic peak position of Raman spectrum (see Appendix A) is 1332.15 cm^−1^, which is the characteristic peak position of the single crystal diamond. Moreover, the NV center is as a deep-level defect in diamond defect with C3v symmetry consisting of a substitutional nitrogen–lattice vacancy pair orientated along the (111) crystalline direction. The center may be found as an ‘in grown’ product of the chemical vapor deposition (CVD) diamond synthesis process added N_2_ gas in bulk and nanocrystalline diamond [37,38]. We investigated the PL spectrum at 325 nm for certifying features of NV^−^ and NV^0^, which are their optical zero phonon lines (ZPLs)at 1.945 eV (637 nm) and 2.156 eV (575 nm), respectively [39]. Thus, PL spectrum of nitrogen-doped diamond films are depicted in Figure 3c. Two peaks around 575 and 625 nm can be discovered in the nitrogen-doped diamond film, indicating ZPL spectral characteristics of the NV^0^ and NV^−^ charge states. Corresponding to Raman spectrum, 340 and 342 nm peaks appear for nitrogen-doped, verifying the existence of diamond and graphite phase one more time, respectively. Besides, we also reveal undoped polycrystal diamond fluorescence properties for contrast, which is illustrated in Figure 3d. A PL peak at 737 nm is attributed to Si–V defect [40]. Few other specific peaks appear except for 340 nm (diamond phase), 342 nm (graphite phase), and 737 nm (Si–V defect). Consequently, the nitrogen-doped diamond film contains extra NV defect energy level in the forbidden gap compared with the undoped diamond film.

The X-ray photoelectron spectroscopy (XPS), an important surface analysis technique, provides surface information with a thickness of 3–5 nm. Here, the XPS is utilized to measure the existence of C-O after experiments, which can strongly identify oxidation reaction on the surface of the diamond. Therefore, the comparison before and after reaction for high-resolution spectra of the C1s is depicted in Figure 4a,b. Simulation of the XPS spectrum toward the surface on the diamond after photocatalyst is shown in Figure 4c,d. the C1s peak was centered on ~283.7 eV [41]. The main oxygenated group in homoepitaxial films is ether (C–O–C), situated at 285.26 eV in {111} films [42,43]. The other oxygenated group is carbonyl (C=O) found at 287.83 eV [44,45,46]. The survey and high-resolution spectra show carbon with evidence of O at the surface after reduction. Thus, we suggested oxide reactions C + H_2_O→CO_2_ + 4H^+^ + 4e^−^ and C + H_2_O→CO + 2H^+^ + 2e^−^ on the surface may be generated in order to keep the law of conservation of electric charge [47,48]. In addition, the XPS survey spectra of nitrogen-doped diamond film after experiments and the high-resolution spectra of the C1s, O1s, and N are measured, as shown in the Appendix A.

N_2_ reduction efficiency with types of the diamond sample under visible and UV illumination (200–800 nm) was investigated. To verify that the nitrogen-doped diamond plays a crucial role in photocatalysis, five different control experiments illustrated in Figure 5a were performed under the same conditions. They are experiments NDD, PD, SCD, Ar gas instead N_2_, silicon, respectively. Moreover, we use the method of obtaining the ammonia production per unit area for comparison due to different substrate growth sizes of the samples. The ammonia yield was calculated by the formula (y = n (nmol)/A (cm^2^)), where A represents reaction area, n was calculated in proportion to the standard value. Standard absorption curve (see Appendix A) was obtained by using ammonium chloride solutions with known concentrations. The original absorption curve tested by UV spectrophotometer is illustrated in Appendix A. The change rate of ammonia production over time of the three diamond samples is shown in Figure 5b. We also performed the linear regression curve fitting by the method of least square, the relevant parameters were presented in Appendix A. Average rates per hour within reaction time of NDD, SCD, and PD were 6.27 ± 1.48 nmol/cm^2^·h, 2.53 ± 0.16 nmol/cm^2^·h, 3.16 ± 0.19 nmol/cm^2^·h, respectively. Obviously, the NDD attributed to the NV center showed the best performance. Grain boundaries in PD resulting in a higher probability of oxygen radical reduced the rate of ammonia production since weak bonds might be dissociated by the irradiation. However, lower efficiency employed by SCD ruled out the influence by CN species that are on the N-doped diamond surface or in the grain boundaries [49,50,51]. Moreover, ammonia yield lower than the calibrated minimum value through Ar instead of N_2_ also confirmed that other species existed on the N-doped diamond surface or in the grain boundaries hardly work in the ammonia production. On the other hand, the almost fixed ammonia yield below the minimum by utilizing silicon precludes the influence of the growth substrate on experiment.

In addition, we also tested the surface resistance (an Agilent B1505A power device analyzer) before and after experiments by sticking the two probes to points 1 cm apart for dozens of times. The average resistance values of nitrogen-doped diamond film changes from 30 KΩ to 100 MΩ, which is consistence with C–O bonds, appeared on the diamond surface after catalysis. Further, it proved hydroxyl terminal may be oxidized to OH•, which may react with carbon atoms to be removed as CO and CO_2_ to maintain charge neutrality.

To explore whether the NV center has an influence on the optical properties of the diamond film, we used an integrating sphere to study the light absorption by converting UV-vis diffuse reflectance spectroscopy (DRS) (see Appendix A) since the incorporation of nitrogen impurities into the diamond film is accompanied by changes in both optical absorption and scattering from the film. The absorbance spectra range from 200 to 800 nm of NDD, PD, SCD are shown in Figure 6**.** Note that there is no vertical offset in the collected spectrum and the maximum saturation absorption of this instrument is 10. Over the entire wavelength region shown, the nitrogen-doped diamond shows higher absorbance, and also exhibits new peaks in the absorption spectrum at approximately ~320 and 460 nm, owing to nitrogen impurities [12,33,34,35]. All of the three samples also show a feature near 220 nm, which corresponds to the absorption for photon energies above the band gap of the diamond (λ < 225 nm). Polycrystalline diamond films show a feature near 240 nm, which is attributed to a direct (momentum-conserving) optical transition of bulk silicon [52,53]. A similar peak on 240 nm in the DRS of silicon (Appendix A) for comparison further proved the conclusion. In addition, the peak on 375 nm appearing in all samples may come from the fact that the deuterium lamp spectrum in this region is somewhat noisy. The results in Figure 5 and the Supporting Information indicated that the presence of nitrogen impurities increases optical absorption in the visible and ultraviolet region.

## 4. Discussion

The mechanism of the influence of nitrogen levels in the diamond for reduction N_2_ to NH_3_ has been studied. The steps can occur by three pathways which are labeled in Figure 7-I, II, and III. The first step (I) in the photocatalyst for reduction indicates intrinsic excitation, the excitation from the conduction band to the valence band, which must occur under deep ultraviolet light. Electron transition from defect levels to the conduction band can be illustrated in the second step (II) and third step (III). The neutral state of NV^0^ (step II), as an acceptor impurity in diamond, can absorb the photon transition from the ground state to the excited state. When the excitation light energy is higher than 2.156 eV, the absorbed photon transitions to the conduction band to become free electrons, thereby breaking away from the bondage of the impurity center. The negatively charged state NV^−^ (step III) transitions from the ground state to the excited state by absorbing an energy greater than zero phonon transition. NV^−^ has a 3A ground state and a 3E excited state, with a 1A state located in between them, to which transitions are (nearly) forbidden, implying that the excited states are close to the band edges, while the ground states are in the middle of the band gap [33,34,35,36,37]. The excitation band is near the conduction band, which represents that it is easier for electrons to jump from nitrogen impurity levels to conduction band. At the same time, conversion from the NV^−^ color center to a neutral NV can be carried out after losing electrons [54]. The intrinsic and extrinsic transitions both occurred under illumination. The radiative transitions between charge states have occurred optically at excited levels via the intermediate valence/conduction band. Furthermore, the non-radiative transitions are believed to connect with tunneling of electrons. When illuminated by light irradiation, the electrons transition would take place at the defect level, simultaneously occurring from valence band to conduction band. Therefore, more photo-generated electrons benefit from the activation of N≡N bonds.

The electrochemical energy scale relative to normal hydrogen electrode (NHE) of the diamond and other band wide gap semiconductors was shown in Figure 8. The photocatalytic reduction of nitrogen requires the following steps. The first step to reduction is to form N_2_H with high energy. Electron transfer reaction H_2_O + e^−^→e^−^ (aq.) has a reduction potential E of −3.2 eV versus NHE, which is the most possible way to form N_2_H [55,56]. In addition, the solvated electrons served as an intermediate to form H• (H^+^ + e^−^→H•). Then, the N_2_H can continuously react with H• to the subsequent reaction with e^−^ and H• to ultimately NH_3_ [57,58,59]. It is notable that the electrochemical potential window of hydrogen terminated diamond is more negative at about 5.2 eV versus NHE. Thus, the diamond can be capable of initiating higher energy reduction reactions. Previous studies reported the approach for reduction of nitrogen was firstly and subsequently studied by using modified forms of TiO_2_ [60,61]. The N_2_ fixation yield in the visible area was less than 67%, due to the re-oxidation of intermediates and products by the pores in the deep valence band of TiO_2_ and weak absorption on the surface of reactant. In addition, the work function of TiO_2_ is 4.6 eV, which means electrons jumped to above the conduction band with barrier [60,61,62,63]. Other wide bandgap semiconductors such as GaN, SiC, due to lower potential energy, have more difficulty for the photocatalyst [56,64,65]. The hydrogen-terminated diamond, currently the only semiconductor with negative electron affinity, was used for an electron state source into liquids for initiating high-energy reduction. This provides a development direction for the field of catalysis.

## 5. Conclusions

Our work demonstrates that nitrogen impurities entering the film can significantly enhance photocatalytic activity, thereby reducing N_2_ to NH_3_. The negatively charged state NV^−^ transitions from the ground state to the excited state by absorbing energy, which contributed to the optical spectra of the films, revealing an increased absorbance both above and below the band gap energy of the diamond. The increase in absorption and scattering results in the diamond’s ability to emit electrons and induce subsequent electron-induced reduction of N_2_ to NH_3_. It is worth mentioning that our method enables the nitrogen fixation reaction to proceed under visible light, thereby reducing the cost of the reaction and promoting its practical application. In this perspective, nitrogen impurity levels in the diamond film have been experimentally proven as the electron source of N_2_ reduction photocatalyst with high catalytic activity.

## Figures and Tables

**Figure 1 materials-13-04559-f001:**
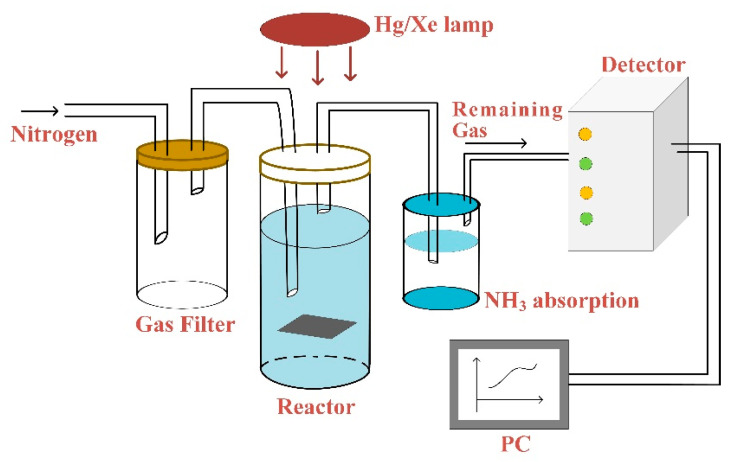
Schematic diagram of simple reaction vessel (Gas filter guarantees purity of reaction gas N_2_ without impurities. Then, N_2_ was slowly bubbled into vessel contained 18.2 MΩ water (Barnsted NanoPure) with 0.01 M high-purity (Alfa Puratronic, 99.9955%) Na_2_SO_4_. The container was sealed by quartz cover and exposed to light for different time periods. The gas exported from vessel was run through the NH_3_ absorption bottle and finally exported from the computer system).

**Figure 2 materials-13-04559-f002:**
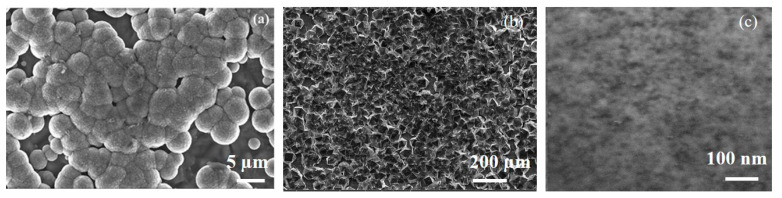
Scanning electron microscope images of diamond film after growth, (**a**) nitrogen diamond film, (**b**) poly-crystalline diamond, (**c**) single crystalline diamond film.

**Figure 3 materials-13-04559-f003:**
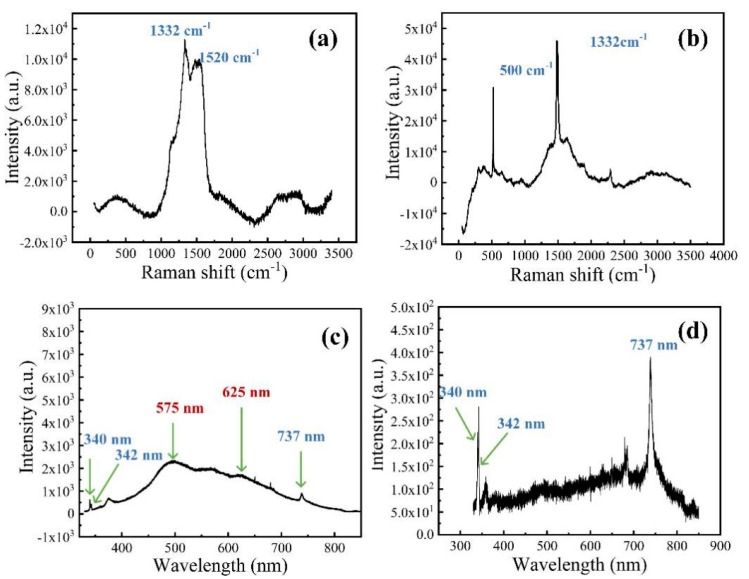
The Raman spectrum of (**a**) nitrogen-doped diamond film, (**b**) polycrystal diamond film. The PL spectrums with excitation at 325 nm of (**c**) nitrogen-doped diamond film and (**d**) polycrystal diamond film after photocatalysis.

**Figure 4 materials-13-04559-f004:**
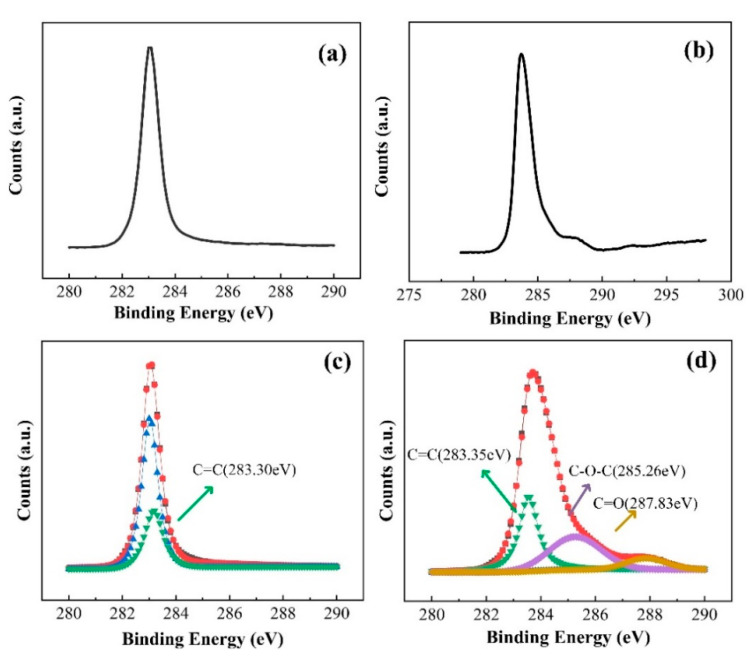
The survey and resolution spectra of C1s. Comparison between before (**a**) and after (**b**) reaction for high-resolution spectra of the C1s. Simulation of XPS spectrum toward surface on diamond before (**c**) and after (**d**) photocatalyst. Carbon–oxygen bonds appeared on the surface after catalysis.

**Figure 5 materials-13-04559-f005:**
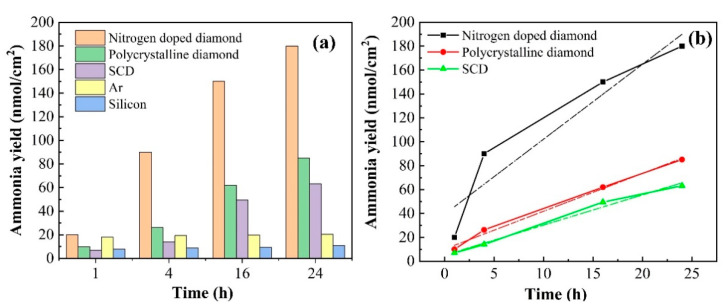
Catalytic efficiency of different control experiments (**a**), experimental curve and linear fit between nitrogen-doped diamond film, SCD film, and polycrystalline diamond film (**b**), the inset is fitting parameters. All of the procedures and experimental instrumentals are identical.

**Figure 6 materials-13-04559-f006:**
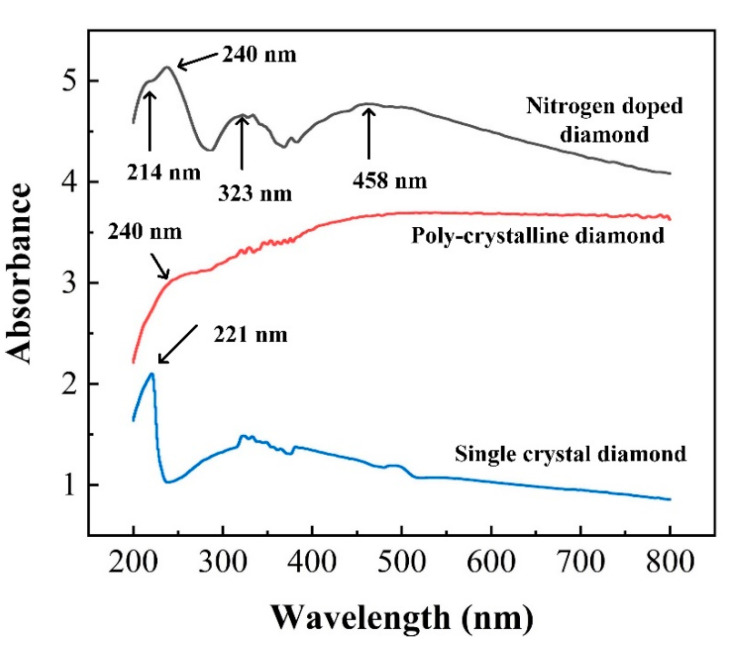
Contrast for absorbance spectra of nitrogen-doped diamond (black line), polycrystalline diamond (red line), and SCD (blue line) in visible and UV regions.

**Figure 7 materials-13-04559-f007:**
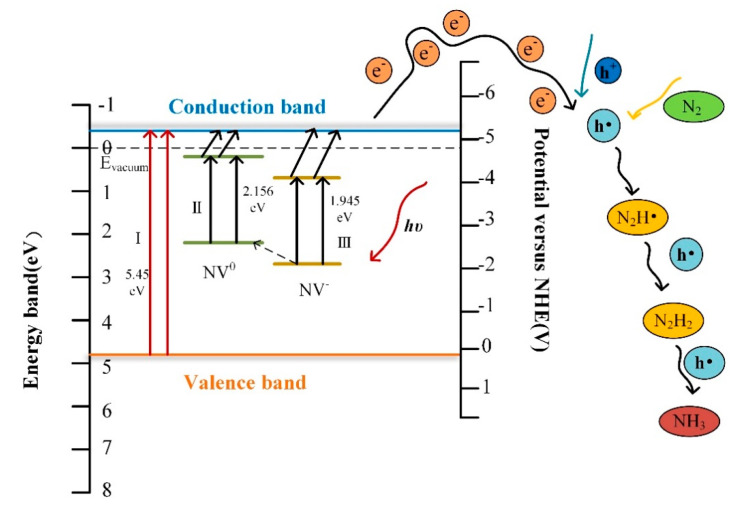
The schematic diagrams of the influence of nitrogen levels on diamond photocatalysis. Step I in the photocatalyst for reduction is intrinsic excitation. The presence of nitrogen levels in diamond can be illustrated in the second step II (the neutral state of NV^0^) and third step III (the negatively charged state NV^−^).

**Figure 8 materials-13-04559-f008:**
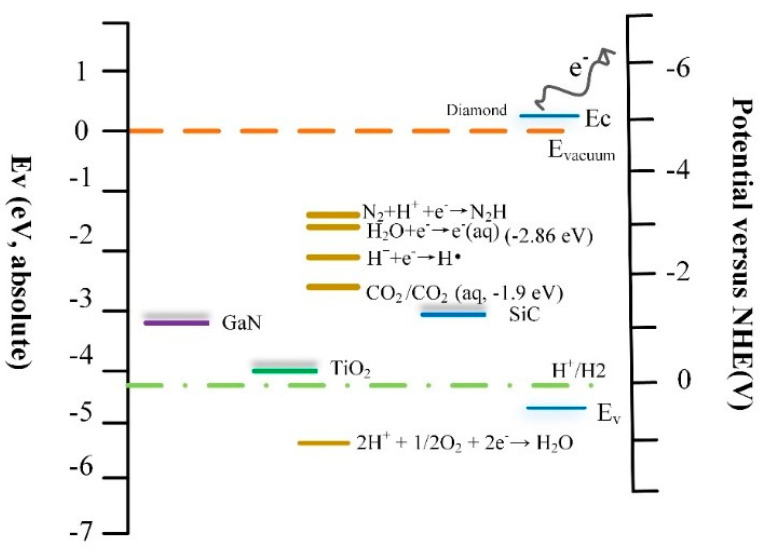
Electrochemical energy scale relative to the NHE of the diamond and other band wide gap semiconductors. Energy scale (left) and the electrochemical energy scale (right) relative to the NHE. Potential for reduction of oxygen to water is shown for pH 7.

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
