# Peer review of "Visible-Light Activation of Photocatalytic for Reduction of Nitrogen to Ammonia by Introducing Impurity Defect Levels into Nanocrystalline Diamond"

_materials, 2020, doi:10.3390/ma13204559_

Round 1

Reviewer 1 Report

The authors have reported the modification of nanocrystalline diamond by introducing N vacancies impurities, and its use as photocatalyst for ammonia reduction process. This is an interesting approach to improve the photocatalytic activity of diamond substrates, which provide certain novelty to this study, although the photocatalytic efficiency reported are very low. Moreover, the careless English language and punctuation used along the manuscript makes some sentences difficult to understand and overshadows the quality and soundness of the report.

There are major issuers that should be addressed before considering this manuscript for publication:

  1. The sentence construction, punctuation and use of English language on this manuscript must be careful revised on its entirety. Here are some examples of confusing sentences:

Lines 30-32: “However, diamond, a promising material for photocatalyst, used for an electron state source into liquids for initiating high-energy reduction reactions such as nitrogen reduction has not been further investigated.” This sentence is contradictory. There are several reports (currently cited by the authors, indeed) on studies of diamond as photocatalyst for ammonia reduction.

Line 41-42: “nitrogen doped diamond film will be investigated as impurities can introduce intermediate levels especially optically active energy levels in diamond forbidden band.”

Line 128-130: “To preclude the impact of CN species that are on the N-doped diamond surface or in the grain boundaries, when then are released and react to form NH3. Single crystalline diamond (SCD) film was studied for a contrast.”

  1. I recommends the authors to improve the figures presentation and quality, especially in figures 4 and 5. In Figure 5b the inset table is unreadable. I suggest presenting it in a supplementary information file.
  1. Despite the modification of the diamond with N vacancies, the obtained photocatalytic efficiency is very low (in the order of nmol). The authors should discuss on the significance of their approach by benchmarking the reported efficiency with related studies.

Minor issues:

  1. I suggest the author to substitute the following expressions with  scientific related vocabulary: Line 28: “As we all know…”; Line 72: “So, …”
  2. All the acronyms must be defined on its first mention. Some examples: Line 60: “NRR”: Line 70: “NCD”; Line 128: “CN”
  3. Line 271: “H2O+e-=e-(aq)” I suggest to check this reaction.

Reviewer 2 Report

The paper shows the features of N-doped diamond as a photocatalyst for N2 reduction to NH3. In their work, the authors present also the performance of   poly-crystalline diamond and a single crystalline diamond film.

The introduction is clear and corresponds to the scope of the research. 

The experimental part sufficiently describes the methodology of materials syntheses and the photoinitiated N2 reduction.

The paper is interesting and deals with important issue of NH3 production via N2 reduction. However, before it is published it must undergo a revision. The main comments are listed below:

1. The SEM observations - It is hard to discuss the difference in materials morphology using SEM pictures displayed in Fig.2. It would be better to show pictures taken at similar zoom. 

Moreover, it seems that the average diameter of N-doped diamond clusters  (Fig. 2a) is significantly bigger than 15nm.

2. Figs. 4 and 5 are of very poor quality and must be improved.

3. The extensive English editing is required owing to a high number of various errors in the text. 

Reviewer 3 Report

This paper describes a potentially useful synthetic approach and includes extensive characterization of the catalytic material. Extensive editing of the English in the text is required, but once completed, this paper will be an interesting and useful contribution to the scientific literature. 

Round 2

Reviewer 1 Report

R. Su and co-authors have well addressed the previous comments, notably improving the quality of the manuscript. I believe that this work should be published as a new contribution to the enhancement of the photocatalytic properties of diamond films.